# Histone Deacetylase Inhibitors and Microtubule Inhibitors Induce Apoptosis in Feline Luminal Mammary Carcinoma Cells

**DOI:** 10.3390/ani11020502

**Published:** 2021-02-15

**Authors:** Filipe Almeida, Andreia Gameiro, Jorge Correia, Fernando Ferreira

**Affiliations:** 1CIISA—Centro de Investigação Interdisciplinar em Sanidade Animal, Faculdade de Medicina Veterinária, Universidade de Lisboa, Avenida da Universidade Técnica, 1300-477 Lisboa, Portugal; filipe.almeida@insa.min-saude.pt (F.A.); agameiro@fmv.ulisboa.pt (A.G.); jcorreia@fmv.ulisboa.pt (J.C.); 2Antiviral Resistance Laboratory, Infectious Diseases Department, National Institute of Health Dr. Ricardo Jorge, Av. Padre Cruz, 1649-016 Lisbon, Portugal

**Keywords:** feline mammary carcinoma, histone deacetylase inhibitors, microtubule inhibitors, therapeutics

## Abstract

**Simple Summary:**

Feline mammary tumors (FMT) are very common in cats, associated with very aggressive behavior and a short life expectancy. Surgery is the most used treatment but tumor recurrence is common. Currently, available therapies are insufficient, therefore, new molecular targets are needed to develop more efficient therapeutics. Histone deacetylases inhibitors (HDACis) have been developed to target tumor cells, by disrupting gene expression and leading to cell death. Microtubules inhibitors (MTIs) have also been a focus of research, to target polymerization of microtubules, and consequently disturbing the cytoskeleton and leading to cell cycle arrest and apoptosis. However, there are few studies on the use of HDACis and MTIs in cats. In this study, we addressed if these two drug classes could be used as new therapeutic options in FMTs. All HDACis and MTIs exhibited suitable and dose-dependent antitumor effects in FMT cell lines. Immunoblot analysis confirmed that the mode of action of HDACis is conserved in feline mammary tumor cell lines. Finally, flow cytometry showed that exposure with HDACis and MTIs lead to the induction of cellular apoptosis. In summary, HDACis and MTIs possess antitumor properties suggesting further studies on their use in the treatment of feline mammary tumors.

**Abstract:**

Feline mammary carcinoma (FMC) is the third most common type of neoplasia in cats, sharing similar epidemiological features with human breast cancer. In humans, histone deacetylases (HDACs) play an important role in the regulation of gene expression, with HDAC inhibitors (HDACis) disrupting gene expression and leading to cell death. In parallel, microtubules inhibitors (MTIs) interfere with the polymerization of microtubules, leading to cell cycle arrest and apoptosis. Although HDACis and MTIs are used in human cancer patients, in cats, data is scarce. In this study, we evaluated the antitumor properties of six HDACis (CI-994, panobinostat, SAHA, SBHA, scriptaid, and trichostatin A) and four MTIs (colchicine, nocodazole, paclitaxel, and vinblastine) using three FMC cell lines (CAT-MT, FMCp, and FMCm), and compared with the human breast cancer cell line (SK-BR-3). HDACis and MTIs exhibited dose-dependent antitumor effects in FMC cell lines, and for all inhibitors, the IC50 values were determined, with one feline cell line showing reduced susceptibility (FMCm). Immunoblot analysis confirmed an increase in the acetylation status of core histone protein HDAC3 and flow cytometry showed that HDACis and MTIs lead to cellular apoptosis. Overall, our study uncovers HDACis and MTIs as promising anti-cancer agents to treat FMCs.

## 1. Introduction

In addition to the clinicopathological characteristics, the molecular immunophenotyping of feline mammary carcinomas (FMCs) is similar to what is described for human breast cancer [1,2]. Moreover, cats share a common environment with humans and therefore exposure to environmental contributors to the development of cancer is similar. Due to these resemblances, the domestic cat has been proposed as a model for comparative oncology [1,3,4,5]. FMCs are predominantly malignant with metastases occurring in most cases, resulting in a poor prognosis and associated with a short life expectancy [6]. Currently, surgery is the most widely used treatment for mammary carcinoma in cats, either alone or in association with chemotherapy. Still, tumor recurrence, even after a full mastectomy, is common [6]. Therefore, new molecular targets are needed to develop more efficient therapeutics.

Histone deacetylases (HDACs) play an important role in the regulation of gene expression at the chromatin level, and their dysregulation has been associated with the development of cancer [7,8]. In recent years, HDACs have been attractive targets for the development of new antitumor agents (HDAC inhibitors—HDACis), which inhibit histone deacetylation activity, leading to chromatin relaxation and uncontrolled expression of essential genes, and consequently, tumor cell cytotoxicity [7,9]. Currently, there are 18 HDAC enzymes described and classified into four classes, according to their homology to yeast HDACs, their subcellular location, their tissue specificity, and their enzymatic activity [7]. So far, at least four HDACis have been approved for human cancer treatment by the US Food and Drug Administration (FDA)—vorinostat (SAHA), romidepsin, belinostat, and panobinostat [8].

In parallel, microtubules (MT) are dynamic structures composed of tubulin polymers that play an important role in cell growth, division, and intracellular trafficking, and have been the focus of research for cancer treatment [10,11]. MT inhibitors (MTIs) interfere with the polymerization of microtubules, causing disruption of the microtubule cytoskeleton, followed by cell cycle arrest and apoptosis [10]. MTIs are classified into two main groups, depending on their effect on microtubule polymers—microtubule-destabilizing agents (colchicine, combretastatins, vinca alkaloyds, cryptophycins, and dolastatins) and the microtubule-stabilizing agents (taxanes, paclitaxel, and epothilones) [11]—and have been in clinical use for the treatment of a large variety of tumors and hematological malignancies [10].

So far, very few studies on the use of HDACis and MTIs have been published in cats [12,13,14]. In this study, we addressed if these two drug classes could be used as new therapeutic options in FMCs. We evaluated the antitumor effect, the mode of action, and cell death mechanism of HDACis and MTIs in three in vitro models of luminal FMC. We found that HDACis and MTIs possess antitumor properties suggesting further studies on their use in the treatment of feline mammary carcinomas.

## 2. Materials and Methods

### 2.1. Feline and Human Cell Lines

The three feline mammary carcinoma cell lines used in this work were CAT-MT (from the American Type Culture Collection; ATCC, Manassas, VA, USA), FMCp, and FMCm (kindly provided by Prof. Nobuo Sasaki, School of Agricultural and Life Sciences, University of Tokyo, Tokyo, Japan). For comparison purposes, we used the human breast cancer cell line SK-BR-3 (from the ATCC, Manassas, VA, USA). CAT-MT and SK-BR-3 cell lines were maintained in Dulbecco’s Modified Eagle Medium (DMEM, Corning, NY, USA,), whereas FMCp and FMCm cell lines were maintained in Roswell Park Memorial Institute 1640 Medium (RPMI, Corning, NY, USA), all supplemented with heat-inactivated 20% (*v*/*v*) fetal bovine serum (FBS; Corning, NY, USA) and incubated at 37 °C in a humidified atmosphere of 5% (*v*/*v*) CO^2^. All cell lines were inspected periodically for their cellular morphology and proliferation rate.

### 2.2. Immunocytochemistry

Expression analysis of estrogen receptor (ER), progesterone receptor (PR), epidermal growth factor receptor type 2 (HER2), cytokeratin 5/6 (CK 5/6), and Ki-67 proliferation index (Ki-67) was performed in feline (CAT-MT, FMCp, and FMCm) and in human (SK-BR-3) cell lines, as previously reported [1]. Briefly, cell lines grown till confluency were embedded in a histogel matrix, enclosed in a paraffin block, and sections of 3 µm thickness were cut and mounted on glass slides (Super Frost Plus, Thermo Scientific, Waltham, MA, USA). Samples were deparaffinized with xylene and rehydrated in distilled water through a series of graded alcohols. The antigen retrieval was conducted in PTLink (DAKO, Carpinteria, CA, USA) for 20 min at 96 °C (for PR, HER2, and Ki-67, citrate buffer at pH 6.0 was used, while for ER and CK 5/6, the tris-EDTA buffer at pH 9.0 was used). Endogenous peroxidase activity was blocked with Peroxidase Block (Novocastra, Leica Biosystems, Milton Keynes, UK) for 15 min, followed by Protein Block (Novocastra, Leica Biosystems, Milton Keynes, UK) for 10 min. Each sample was incubated with the primary antibody for 1 h in a humidified chamber (ER—clone EP1, PR—clone 1E2, CK 5/6—clone D5/16B4, all ready-to-use from Ventana, Roche, Switzerland; and Ki-67—clone MIB-1, 1:200 from DAKO, USA), or overnight at 4 °C (HER2—clone CB11, 1:200 from Abcam, Cambridge, UK). After 30 min in Post-Primary solution (Novocastra, Leica Biosystems, UK), samples were incubated with 3,3′-diaminobenzidine (DAB) solution (Novocastra, Leica Biosystems, UK). Then, cells were counterstained with Mayer’s hematoxylin (Merck, NJ, USA) and mounted in Entellan (Merck, Darmstadt, Germany). All samples were evaluated by a pathologist and scored using the American Society of Clinical Oncology’s guidelines, as summarized in Appendix A [1,15].

### 2.3. Cytotoxicity Assays

To investigate the effect of six HDACis (CI-994, panobinostat, SAHA, SBHA, scriptaid, and trichostatin A; EPI009, Sigma-Aldrich, Merck, Darmstadt, Germany) and four MTIs (colchicine, nocodazole, paclitaxel, and vinblastine; kindly provided by Dr. Frederico Aires da Silva, FMV-ULisboa, Lisbon, Portugal), cell viability assays were performed using the Cell Proliferation Reagent WST-1 (Abcam, Cambridge, UK), according to the manufacturer’s instructions. Briefly, the feline cell lines CAT-MT, FMCp and FMCm, and the human cell line SK-BR-3 were seeded in 96 well plates to obtain confluency of 90% after 24 h (5 × 10^3^ cells/well for CAT-MT and FMCp, 1.5 × 10^4^ cells/well for FMCm, and 1 × 10^4^ cells/well for SK-BR-3). Then, cells were left unexposed or exposed to increasing concentrations of each HDACi (1 μM to 250 μM for CI-994, 25 nM to 5 μM for panobinostat, 250 nM to 50 μM for SAHA, 10 μM to 1000 μM for SBHA, 250 nM to 50 μM for scriptaid, and 250 nM to 50 μM for trichostatin A) or MTIs (0.05 nM to 1 μM for colchicine, 0.5 nM to 2 μM for nocodazole, 0.5 nM to 2 μM for paclitaxel, and 0.5 nM to 2 μM for vinblastine). Dimethyl sulfoxide, knowns as DMSO (for CI-994, panobinostat, SAHA, SBHA, scriptaid, trichostatin A, nocodazole, paclitaxel, and vinblastine) and ethanol (for colchicine) were used as vehicle control at a maximum final concentration of 1%. After 72 h of drug exposure, the WST-1 reagent was added, followed by an incubation period of 4 h at 37 °C. Absorbance at 440 nm was measured using a plate reader (FLUOStar Optima, BMG LABTECH GmbH, Ortenberg, Germany). Duplicate wells were used to determine each data point and three independent experiments were performed on different days. Best-fit IC50 values were calculated using the log (inhibitor) vs. response (variable slope) function in GraphPad Prism (version 5.00 for Windows, GraphPad Software, San Diego, CA, USA).

### 2.4. Immunoblotting

To prepare total cell protein extracts, the feline cell lines CAT-MT and FMCp and the human cell line SK-BR-3 were seeded in 24 well plates (two wells per sample) in order to obtain confluency of 90% after 24 h (5 × 10^4^ cells/well for CAT-MT and FMCp, 1 × 10^5^ cells/well for FMCm, and 7 × 10^4^ cells/well for SK-BR-3). Then, cells were left unexposed or exposed to the IC50 concentration of each HDACi. DMSO was used as vehicle control at a maximum final concentration of 1%. After 72 h of drug exposure, cells were collected using 50 μL of RIPA lysis buffer (50 mM TrisHCL pH 8.0, 150 mM NaCl, 1% Tween-20, 0.5% sodium deoxycholate, 0.1% SDS, 5 mM EDTA, pH 8.0) supplemented with 1% 100× Halt™ protease inhibitor cocktail EDTA-free (Thermo Fischer, Waltham, MA, USA) and 2 mg/mL Iodoacetamide (AppliChem, Darmstadt, Germany), and stored at −80 °C. Before immunoblotting, total cell protein extracts were resuspended in 10 μL of SDS-PAGE Loading Buffer 5× (NZYTech, Lisbon, Portugal), denatured 10 min at 100 °C, and resolved in 12% SDS-PAGE. Proteins in the gel were transferred onto nitrocellulose membranes (Amersham Protran 0.45 NC, GE Healthcare, Chicago, IL, USA) and blocked in 4% (*w*/*v*) dried skimmed milk diluted in PBS containing 0.1% (*v*/*v*) Tween-20. The membranes were probed with primary and horseradish peroxidase-conjugated secondary antibodies (rabbit polyclonal anti-acetyl-Histone H3 [Lys9/Lys14], 1:1000, Cell Signaling Technologies, MA, USA; mouse monoclonal anti-β-actin [AC-15], Abcam, Cambridge, UK; Goat HRP anti-rabbit IgG H&L, Abcam, UK; goat HRP anti-mouse IgG H&L, Abcam, UK) and detected using ECL Clarity (Bio-RAD, Hercules, CA, USA), in a Chemidoc XRS+ System (Bio-Rad, Hercules, CA, USA). Three independent experiments were performed on different days. Densitometry of the correspondent bands was performed using Fiji Image J (open-source version for Windows^®^) and statistical analyses were conducted using GraphPad Prism (version 5.00 for Windows^®^, GraphPad Software, San Diego, CA, USA). The *p*-values were calculated by a two-tailed unpaired Student’s t-test relative to non-treated cells (ns—*p* > 0.05, *—*p* ≤ 0.05, **—*p* ≤ 0.01, ***—*p* ≤ 0.001).

### 2.5. Flow Cytometry

The percentage of apoptotic cells after HDACis or MTIs exposure was measured by using the Guava Nexin Assay (Merck, Darmstadt, Germany), according to the manufacturer’s instructions. The different staining of the cells allows to distinguish between viable (Annexin V/7-AAD double negative), early apoptotic (Annexin V positive/7-AAD negative), and late-apoptotic/dead cells (Annexin V/7-AAD double positive). The feline cell lines CAT-MT and FMCp, and the human cell line SK-BR-3 were seeded in 24 well plates (two wells per condition) to obtain confluency of 90% after 24 h (5 × 10^4^ cells/well for CAT-MT and FMCp, 1 × 10^5^ cells/well for FMCm, and 7 × 10^4^ cells/well for SK-BR-3). Cells were then left unexposed or exposed to the IC50 concentration of each HDACi. DMSO (CI-994, panobinostat, SAHA, SBHA, scriptaid, trichostatin A, nocodazole, paclitaxel, and vinblastine) and ethanol (colchicine) were used as vehicle control at a maximum final concentration of 1%. After 72 h of exposure, supernatants were harvested and the remaining attached cells were trypsinized and added to the correspondent supernatants. Samples were centrifuged for 5 min at 500 g and at room temperature, washed with PBS, and resuspended in 500 μL PBS containing 2% FBS. To each sample, 50 μL Guava Nexin reagent was added and incubated for 20 min at room temperature (protected from light), before acquisition in a BD LSR Fortessa ×20 (at Fundação Champalimaud). Data were analyzed using FlowJo™ v10.6.1 (BD, San Jose, CA, USA).

## 3. Results

### 3.1. CAT-MT, FMCp, and FMCm Are Feline Luminal Mammary Carcinoma Cell Lines

To determine if the used feline tumor cell lines could be a valuable model to screen for therapeutic agents for the treatment of feline mammary carcinomas, the three lines were immunophenotyped. While the CAT-MT cell line derives from a highly malignant tubular adenocarcinoma [16], the FMCp and FMCm cell lines were originated from an adenocarcinoma, collected from a primary and a lymph node metastatic lesion, respectively, from the same diseased cat [17]. The human SK-BR-3 cell line was selected for comparison purposes and derives from an adenocarcinoma collected from a metastatic site [18]. The expressions of the estrogen receptor (ER), progesterone receptor (PR), feline and human epidermal growth factor receptor type 2 (HER2), cytokeratin 5/6 (CK 5/6), and proliferation index (Ki-67) were evaluated in all cell lines and the results obtained are depicted in Figure 1 and Table 1. For the molecular classification of the cell lines, the guidelines of the St. Gallen International Expert Consensus panel were followed, as previously described [1]. Immunostaining analysis revealed that the CAT-MT cell line has a Luminal A subtype, FMCp and FMCm cell lines have a Luminal B subtype, and the human breast cancer cell line has a HER2+ subtype. These results suggest that CAT-MT, FMCp, and FMCm cell lines are suitable models to screen for therapeutic agents for feline luminal A and B mammary carcinoma subtypes.

### 3.2. HDACis and MTIs Present Antitumor Effects on Luminal Feline Cell Lines CAT-MT and FMCp

To evaluate the potential use of HDACis and MTIs in luminal feline mammary carcinoma, we evaluated and compared the antitumor effects of a panel of six HDACis (CI-994, panobinostat, SAHA, SBHA, scriptaid, and trichostatin A) and four MTIs (colchicine, nocodazole, paclitaxel, and vinblastine), using the three above mentioned feline cell lines and the human breast cancer cell line, as control. CAT-MT, FMCp, FMCm, and SK-BR-3 were incubated with increasing concentrations of each HDACi or MTIs for 72 h and the cytotoxicity of the compounds was measured using the WST-1 assay. As shown in Figure 2, all tested HDACis (Figure 2A) and MTIs (Figure 2B) exhibited dose-dependent antitumor effects on FMC cell lines, and on the human cell line SK-BR-3. However, for the feline cell line FMCm, we observed a pattern of reduced susceptibility to almost all drugs tested (black lines in Figure 2). No evident cytotoxicity was detected for vehicle-only (DMSO or ethanol) exposed cells (data not shown). The data obtained allowed us to calculate the half-maximal inhibitory concentration (IC50) for the feline cell lines CAT-MT and FMCp and for the human cell line SK-BR-3. However, in a few cases (panobinostat and trichostatin A for FMCp, and paclitaxel for SK-BR-3), data were not sufficient to calculate the IC50. Table 2 shows the best-fit IC50 values of each HDACi (µM range) and MTI (nM range), using the log (inhibitor) versus response (variable slope) function. The IC50 concentrations range from 0.015 to 62.980 µM for HDACis, and from 0.570 nM to 236,200 nM for MTIs. Of the HDACis, panobinostat presented the higher cytotoxic activity (IC50 = 0.042 ± 0.067 µM for CAT-MT and IC50 = 0.015 ± 0.027 µM for SK-BR-3), followed by trichostatin A (IC50 = 0.263 ± 0.062 µM for CAT-MT and IC50 = 0.572 ± 0.029 µM for SK-BR-3). Of the MTIs, colchicine presented the higher cytotoxic activity (IC50 = 1.472 ± 0.484 nM for CAT-MT, IC50 = 5.876 ± 0.968 nM for FMCp, and IC50 = 0.590 ± 0.350 nM for SK-BR-3). Although the remaining HDACis and MTIs presented lower cytotoxicity effects in both feline and human cell lines, the results obtained strongly suggest that some HDACis and MTIs present antitumor effects on luminal feline carcinoma cell lines CAT-MT and FMCp.

### 3.3. The Cytotoxic Effect of HDACis Is Related to Histone Acetylation

The main mode of action of the HDACi compounds is to modify the acetylation status of core histone proteins. The inhibition of histone deacetylation enzymes results in the accumulation of the acetylated forms inside the cell nucleus, leading to chromatin relaxation and subsequent cytotoxicity due to disrupted gene expression. To understand if HDACis are inhibiting histone deacetylation in the feline cell lines, we evaluated the acetylation status of the histone H3 by immunoblot and densitometry. CAT-MT, FMCp, and SK-BR-3 cell lines were incubated with the correspondent IC25 and IC50 of each compound for 72 h (Table 2), and total cell protein extracts were collected and analyzed. Figure 3A shows a representative immunoblot image demonstrating higher levels of acetylated histone H3 (Lys9/Lys14), after treatment of the different HDACi compounds. Densitometry analysis (Figure 2B) demonstrated that the feline cell lines, in agreement with the human cell line and in comparison to the unexposed cells, presented higher levels of acetylated histone H3 (Lys9/Lys14) after 72 h of exposure with each HDACi. Only for trichostatin A, the differences found between exposed and unexposed cells were not significant (calculated p-values depicted in the graph). Altogether, the results show that exposure of the feline cell lines CAT-MT and FMCp to HDACis leads to an increase in the levels of histone H3 acetylated on lysine-9 and -14, indicating that the mechanism of action of HDACis is conserved in feline mammary carcinoma cell lines.

### 3.4. Apoptosis Is the Main Mechanism by Which HDACis and MTIs Induce Tumor Cell Death

To confirm that the reduction in cell viability of the feline cell lines after exposure with HDACi or MTI compounds is consistent with the induction of apoptosis, we analyzed Annexin V/7-AAD staining by flow cytometry, following 72 h of incubation with the IC50 concentration of each HDACi or MTI (Table 2). As shown in Figure 4, we found a higher percentage of cells in early or late apoptosis, when comparing unexposed cells with HDACi- or MTI-exposed cells. For CAT-MT, although in non-treated cells there were about 50% apoptotic cells (27% in early apoptosis and 23% in late apoptosis; partially explained by an excessive cell growth), in HDACi-treated cells the percentage of cells in late apoptosis was higher, and varied from 27.4% to 64.2% (SAHA and panobinostat treatments, respectively), and in MTIs between 25.5% and 46.3% (nocodazole and paclitaxel treatments, respectively). We also observed that in MTI-treated cells, the percentage of cells in early apoptosis was slightly higher (on average 30%, compared to less than 10% in HDACi-treated cells). For FMCp, we observed for HDACis drugs, an average of 50% cells in late apoptosis, while for non-treated cells that value was less than 5%. However, in MTI-treated cells, the percentage of cells in early apoptosis was higher (average of ~35%) than the percentage of cells in late apoptosis (average of ~5%). For SK-BR-3, the results were more consistent and showed that on average, 60% of exposed cells were in late apoptosis (the percentage of cells in early apoptosis was on average 10%). Overall, these results are in agreement with the cell viability results, indicating that the cytotoxic activity of HDACis or MTIs in the feline cell lines is consistent with the induction of apoptosis.

## 4. Discussion

Because the majority of FMTs are malignant, early detection and effective therapy have a major impact on survival time [6]. Thus, the development of more efficient therapeutics is of most importance. In this study, we evaluated the antitumor properties of histone deacetylase inhibitors (HDACis) and microtubule inhibitors (MTIs) using in vitro models of luminal feline mammary carcinomas. We found that the cytotoxic effect, the mode of action, and induction of apoptosis of HDACis and MTIs are conserved between the feline mammary carcinoma cell lines (CAT-MT and FMCp) and a human breast cancer cell line (SK-BR-3). At least four HDACis have already been approved for cancer treatment by the US Food and Drug Administration (FDA)—vorinostat (SAHA), romidepsin, belinostat, and panobinostat [7]. Altogether, these results suggest HDACis and MTIs as novel antitumor agents for the treatment of feline mammary carcinomas and emphasize the role of the domestic cat as a model for comparative oncology.

New drugs for breast cancer treatments have been investigated mainly in murine cancer models, such as genetically engineered mice or human-mouse xenograft models. Although their unquestionable value, they have some limitations [5,19]. While in humans, as well as in cats and dogs, the tumors arise spontaneously, in the mouse models, tumors have to be induced. Furthermore, there are biological differences between spontaneously occurring cancers and transplanted cancers in mice that can lead to divergences in carcinogenesis due to different tumor development and variation in activated genes and pathways [20]. In this study, we used three feline luminal mammary carcinoma cell lines as in vitro models for screening novel antitumor drugs for the treatment of feline mammary carcinomas. Although the doubt remains whether cell lines fully represent or not the original tumor, there is a high degree of genomic similarity between the original tumor and their derived cancer cell lines [5,21]. Furthermore, cell lines derived from a patient could represent the first line of research to understand the biology of the tumor and to predict therapy effectiveness. For the feline counterpart, however, the number of available tumor cell lines is still limited, existing only about five studies describing feline mammary carcinomas and the cell lines derived from them [16,17,22,23,24]. We found that the feline cell lines CAT-MT, FMCp, and FMCm are representative of luminal A and B feline mammary carcinomas. Moreover, studies performed in our lab, involving a large number of female cats, demonstrated the molecular heterogeneity of feline mammary carcinomas and revealed that Luminal A and B are common types in both cats and humans [1,2]. Therefore, the cell lines used in this study are valuable in vitro models to screen for therapeutic agents for the treatment of feline mammary carcinomas.

In this study, both HDCAis and MTIs showed promising results in CAT-MT and FMCp cell lines. However, the FMCm cell line exhibited a pattern of reduced susceptibility when exposed to these compounds. Indeed, the acetylation levels of histone H3 remain unchanged in the FMCm cell line, after exposure to the IC50 values obtained for CAT-MT (data not shown). Drug resistance has a dramatic effect on cancer therapy and a drastic impact on patient survival. Drug resistance can develop at the cell level (changes in the target protein) or as a consequence of poor accessibility of the compound (changes in the transport in and out of the cell) [25]. Multiple drug resistance is often associated with increased drug efflux from the cells mediated by energy-dependent transporters. The most common and best-studied phenotype is the overexpression of one member of the adenosine triphosphate (ATP) binding cassette family of transporters, the P-glycoprotein [26]. Many common antitumor agents, as vinblastine, doxorubicin, and paclitaxel are substrates of this transporter [25]. We analyzed by immunoblotting the overexpression of P-glycoprotein in all the cell lines tested from samples collected either in the presence or absence of HDACis or MTIs but failed to detect the protein (data not shown).

Regarding the veterinary clinical setting, the therapeutic potential of HDACis have been demonstrated, but mainly on dogs. Vorinostat (SAHA) was evaluated in canine cell lines and corresponding xenograft models and showed promising results in the treatment of relevant dog cancers, such as canine urothelial carcinoma, osteosarcoma, prostatic carcinoma, T-cell lymphoma, and melanoma [27,28]. The same panel of HDACis used in our study was evaluated in canine lymphoma, as a comparative model for non-Hodgkin lymphoma [29]. The authors were able to show that panobinostat efficiently inhibited mouse-induced xenograft tumor growth, by promoting acetylation and apoptosis in vivo. In our study, panobinostat also presented significant results, corroborating our work and its use in human and veterinary medicine. Another HDACi tested, AR-42, a phenylbutyrate-based class I/IIB HDAC inhibitor, was revealed as being more potent than SAHA for the treatment of both canine and human osteosarcoma cancer [30]. Moreover, the same authors show a synergistic effect between AR-42 and doxorubicin, suggesting here the valuable use of combined therapy protocols. Considering the MTIs, they have been studied for a long time in dogs, as the example of vinblastine and paclitaxel [31,32]. However, as multiple drug resistance is common to occur with this class of drugs, new candidates have been constantly characterized [31]. Other compounds or classes of compounds have been evaluated and are currently in use in human medicine and dogs, but have limited application in domestic cats, such as the small-molecule tyrosine kinase (TK) inhibitors [33]. cat. Recently, bevacizumab, which targets the vascular endothelial growth factor and is a key regulator of tumor angiogenesis, proliferation, and metastasis, was shown to suppress tumor growth in a xenograft model of feline mammary tumor [34]. Drugs that target epigenetic alterations, such as the DNA methyltransferase inhibitor 5-azacytidine (5-AzaC), were already approved for the treatment of hematological malignancies and are currently being applied in breast cancer. Treatment with a 5-AzaC was found to be toxic to tumors, but not to healthy mammary cell lines from dogs, cats, and humans, validating the therapeutic potential of this drug [35]. Another target receiving attention is the peptidyl arginine deiminase enzymes (PAD), which is involved in citrullination. It was found that the PAD inhibitor BB-CLA is capable of reducing the viability of canine and feline mammary cancer cell lines with minimal effects on normal mammary cells [36]. The development of new therapeutic options for FMC is a growing field with innumerous possibilities. Particularly, in this study, we found that HDACi and MTI compounds are great candidates for the treatment of luminal B FMCs, although further studies are needed to evaluate their effects in domestic cats.

## 5. Conclusions

This study tested six HDACis and four MTIs in feline mammary carcinoma cell lines, demonstrating its antitumor effect, and also a conserved genetic and cell death mechanisms, by comparison with the human cell line, SK-BR-3. Moreover, this study suggests new molecular targets for the treatment of mammary tumors in cats, and extend the knowledge about the similarities between the feline mammary tumor and human breast cancer, further supporting the utility of the domestic cat as a valuable model for comparative oncology studies.

## Figures and Tables

**Figure 1 animals-11-00502-f001:**
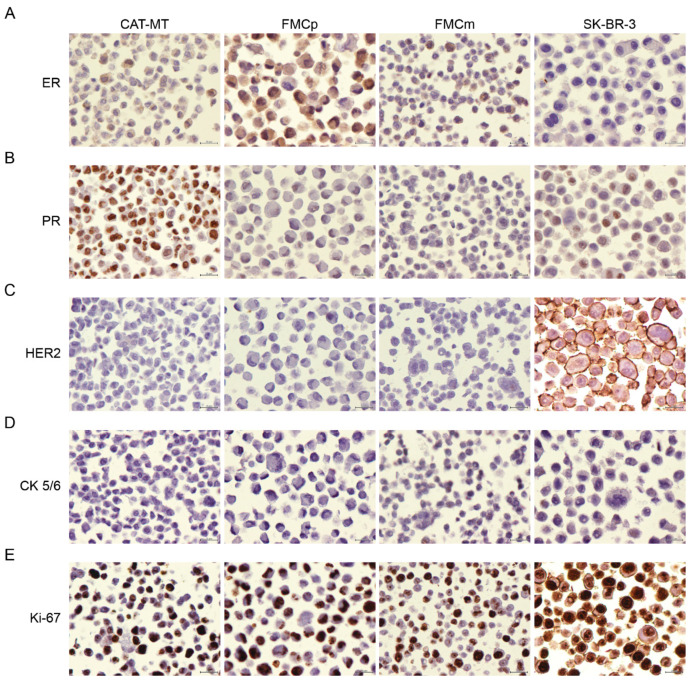
Molecular subtypes of feline mammary carcinoma cell lines. The feline cell lines (CAT-MT, FMCp, and FMCm) and the human breast cancer cell line SK-BR-3 were evaluated by immunocytochemistry for (**A**) ER, (**B**) PR, (**C**) HER2, (**D**) CK 5/6, and (**E**) proliferation index (Ki-67). CAT-MT shows weak (5%) staining for ER, strong (70%) staining for PR, negative staining for HER2, negative staining for CK 5/6, and strong (~50%) staining for Ki-67. FMCp shows strong (~70%) staining for ER, negative staining for PR, HER2, and CK 5/6, and strong (~60%) staining for Ki-67. FMCm shows weak (10%) staining for ER, negative staining for PR, HER2, and CK 5/6, and strong (~70%) staining for Ki-67. SK-BR-3 shows negative staining for ER, average (3%) staining for PR, intense staining for HER2, negative staining for CK 5/6, and strong (~50%) staining for Ki-67. Original magnification ×400 (Mayer’s hematoxylin).

**Figure 2 animals-11-00502-f002:**
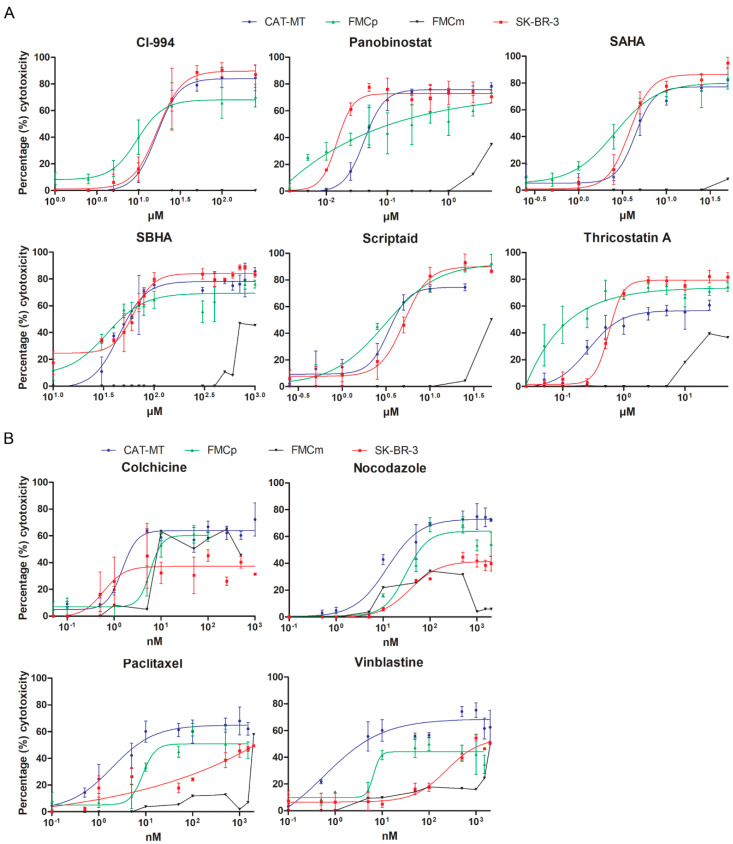
Histone deacetylases inhibitors (HDACis) and microtubules inhibitors (MTIs) present cytotoxicity effect on feline cell lines CAT-MT and FMCp. The feline cell lines (CAT-MT, FMCp, and FMCm) and the human breast cancer cell line SK-BR-3 were incubated with increasing concentrations of (**A**) HDACis (CI-994, panobinostat, SAHA, SBHA, scriptaid, and trichostatin A and (**B**) MTIs (colchicine, nocodazole, paclitaxel, and vinblastine). After 72 h, cell viability was measured using WST-1 (abcam). The log (inhibitor) vs. response (variable slope) function in Graph Pad Prism (version 5.00 for Windows^®^, GraphPad Software, San Diego, CA, USA) used to calculate the IC50 value is shown for CAT-MT, FMCp, and SK-BR-3 cell lines. Data was not sufficient to calculate IC50 values for FMCm cell line. DMSO (CI-994, panobinostat, SBHA, SAHA, scriptaid, trichostatin A, nocodazole, paclitaxel, and vinblastine) and ethanol (colchicine) were used as vehicle control.

**Figure 3 animals-11-00502-f003:**
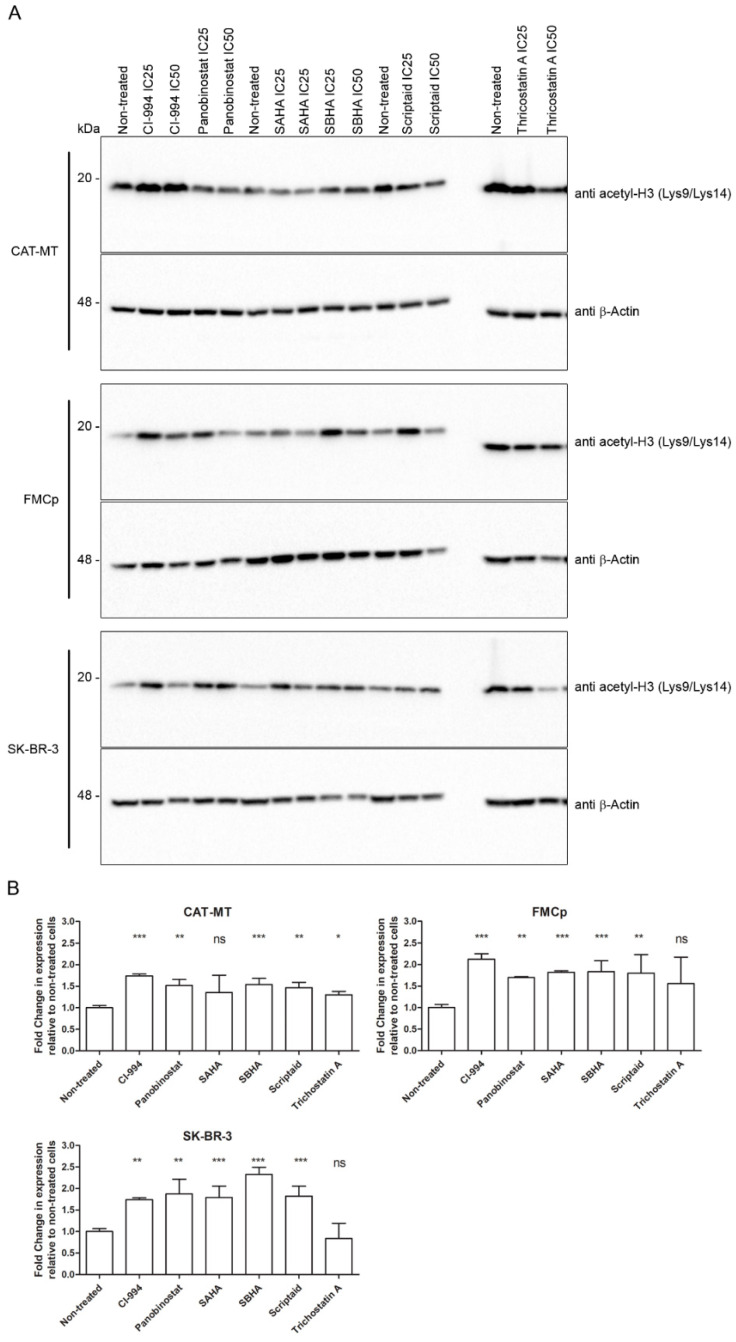
The cytotoxic effect of HDACis correlates with the acetylation status of the histone H3 (Lys9/Lys14). CAT-MT, FMCp, and SK-BR-3 cell lines were exposed to the correspondent IC50 of each HDACi for 72 h (for panobinostat and trichostatin A in the FMCp cell line, the IC50 values used were the ones obtained for CAT-MT). Cells were collected for total cell protein extracts and the levels of acetylation were assessed by (**A**) immunoblot with anti-acetyl-Histone H3 [Lys9/Lys14] antibody (Cell Signaling Technologies, Danvers, MA, USA) and (**B**) densitometry and compared to unexposed cells. DMSO (CI-994, panobinostat, SBHA, SAHA, scriptaid, and trichostatin (**A**) was used as vehicle control and sample loading was controlled with β-actin antibody (AC-15, Abcam, Cambridge, UK). In (**A**), a representative immunoblot is shown. Statistical analysis was performed in Graph Pad Prism software and *p*-values were calculated by a two-tailed unpaired Student’s t-test relative to non-treated cells (ns *p* > 0.05, * *p* ≤ 0.05, ** *p* ≤ 0.01, *** *p* ≤ 0.001).

**Figure 4 animals-11-00502-f004:**
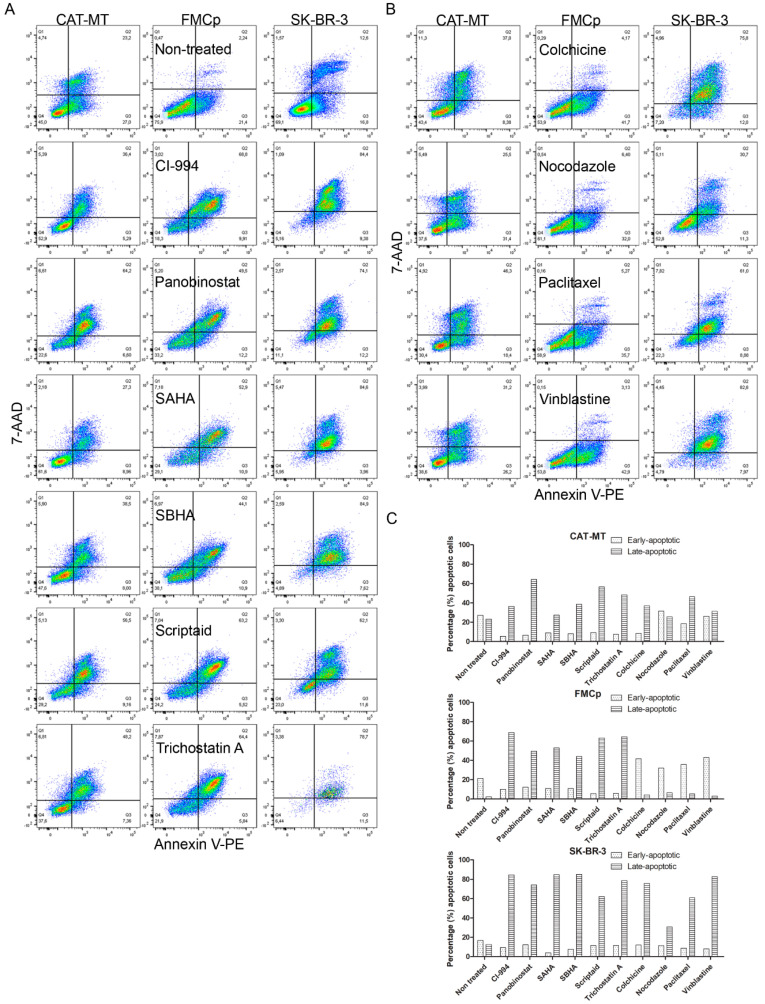
HDACis and MTIs induce apoptosis in feline cell lines. CAT-MT, FMCp, and SK-BR-3 cell lines were incubated with the correspondent IC50 of each HDACi (**A**) or MTI (**B**) for 72 h (for panobinostat and trichostatin A in the FMCp cell line and for paclitaxel in the SK-BR-3 cell line, the IC50 values used were the ones obtained for CAT-MT). Cells were collected, stained with Guava Nexin, and subjected to flow cytometry. (**C**) The percentage of apoptotic cells was determined according to Annexin V/7-AAD staining.

**Table 1 animals-11-00502-t001:** Molecular subtypes of feline mammary carcinoma cell lines. The feline cell lines (CAT-MT, FMCp, and FMCm) and the human breast cancer cell line SK-BR-3 were evaluated by immunocytochemistry for estrogen receptor (ER), progesterone receptor (PR), epidermal growth factor receptor type 2 (HER2), cytokeratin 5/6 (CK 5/6), and Ki-67 molecular markers. The scoring criteria are presented in Appendix A.

Cell Line	CAT-MT		FMCp		FMCm		SK-BR-3	
	Score	Obs.	Score	Obs.	Score	Obs.	Score	Obs.
ER	3	5% (Weak)	8	70% (Strong)	3	10% (Weak)	0	No signal
PR	8	80% (Strong)	0	No signal	0	No signal	4	3% (Average)
HER2	0	Negative	0	Negative	0	Negative	+3	Intense
CK 5/6	Negative	<1%	Negative	<1%	Negative	<1%	Negative	<1%
Ki-67	Strong	50.2%	Strong	57.4%	Strong	68.5%	Strong	53.6%
Molecular Subtype	Luminal A		Luminal B		Luminal B		HER2+	

**Table 2 animals-11-00502-t002:** Best-fit IC50 values (Mean and SEM) of each HDACi and MTI, calculated using the log (inhibitor) versus response (variable slope) function in Graph Pad Prism software.

Drug	Cell Line	CAT-MT	FMCp	FMCm	SK-BR-3	IC50 Units
HDACis	CI-994	16.470 (±1.904)	9.616 (±2.150)	ND	16.860 (±3.183)	μM
	Panobinostat	0.042 (±0.067)	ND	ND	0.015 (±0.027)	
	SAHA	4.416 (±0.453)	2.571 (±0.578)	ND	3.798 (±0.344)	
	SBHA	45.230 (±4.692)	33.830 (±6.454)	ND	62.980 (±4.033)	
	Scriptaid	3.392 (±0.403)	3.090 (±0.691)	ND	5.095 (±0.487)	
	Trichostatin A	0.263 (±0.062)	ND	ND	0.572 (±0.029)	
MTIs	Colchicine	1.472 (±0.484)	5.876 (±0.968)	ND	0.590 (±0.350)	nM
	Nocodazole	12.270 (±3.455)	30.840 (±8.499)	ND	37.120 (±8.844)	
	Paclitaxel	1.939 (±1.134)	8.646 (±2.337)	ND	ND	
	Vinblastine	0.570 (±1.080)	6.563 (±1.514)	ND	236.200 (±91.868)	

ND—Not determined for panobinostat and trichostatin A on FMCp, and paclitaxel on SK-BR-3, where the IC50 values used were the ones obtained for CAT-MT.

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
