# Peer review of "Histone Deacetylase Inhibitors and Microtubule Inhibitors Induce Apoptosis in Feline Luminal Mammary Carcinoma Cells"

_animals, 2021, doi:10.3390/ani11020502_

Round 1
Reviewer 1 Report
Authors Almeida, et al. present an interesting study to understand the effects of HDAC and microtubule-inhibitors in feline luminal mammary carcinoma cells. In the form this paper was presented, I found it very easy to read. The results were presented very clearly and I have no specific critiques for anything that is presented. It would definitely be interesting if the feline cell lines could be implanted in alternative animal models (e.g. mice) as xenografts and then therapies be tested to provide a more convincing in vivo validation of the findings.
Author Response
Reviewer 1
“Authors Almeida, et al. present an interesting study to understand the effects of HDAC and microtubule-inhibitors in feline luminal mammary carcinoma cells. In the form this paper was presented, I found it very easy to read. The results were presented very clearly and I have no specific critiques for anything that is presented. It would definitely be interesting if the feline cell lines could be implanted in alternative animal models (e.g. mice) as xenografts and then therapies be tested to provide a more convincing in vivo validation of the findings.”
Dear reviewer, thank you so much for reviewing our manuscript and for your positive remarks on our work. In response to your final observation, it is something that we want to revisit in the near future. Thanks for the interesting suggestion.
Reviewer 2 Report
Review of manuscript: “Characterization of antitumor effects of histone deacetylase- and microtubule-inhibitors in feline luminal mammary carcinoma cells”
In the present study, it was done a screening of histone deacetylases inhibitors (HDACi) and microtubules inhibitors (MTi) in feline mammary tumor cell lines. Pharmacological treatment options against neoplasia in pets are scarce. The work is relevant and important to the knowledge area. However, the manuscript requires attention.
1- I guess that the title needs be rewritten. It is not representing the work. The work is a screening of HDACi and MTi in luminal mammary cancer cells. It is not a characterization of antitumor effects of those compounds. I suggest “Histone deacetylase inhibitors and microtubule inhibitors induce apoptosis in feline luminal mammary carcinoma cells”
2- Introduction section, page 2, line 59. “HDACs have been attractive targets for…”; instead HDAC have been attractive candidates for…
3- Materials and Methods section, page 2. Did the authors perform any procedure to evaluate if the cells were contaminated with mycoplasma?
4- Materials and Methods section, page 4. Why did the authors use Histone H3 [Lys9/Lys14] as a target for acetylation? Others lysine residues as K27 (H3K27) or even other histones as H4 are important targets of HDACi in cancer control.
5- Materials and Methods section, page 4. β-actin is a constitutive cytoplasmatic protein. Why did the authors use β-actin as western blot control for histone acetylation?
6- Page 5, table 1. Table 1 must be reformatted. It is confused.
7- Page 8, table 2. Table 2 must be reformatted.
8- Page 8, subtitle “The cytotoxic effect of HDACi is correlated with epigenetic modifications” This is not true. The authors did not perform statistical analysis that support it.
9- Figure 3. The representative image of the western blot must be replaced. The image must represent all bands obtained in immunoblotting, without cuts or selected bands. A representative image is required for each cell line. And each graph must have properly indicated the legend of the ordinate and abscissae axis. Why did the authors analyze the results with two-tailed unpaired Student’s t-test relative to non-treated cells? Are not differences between HDACis important?
10- Discussion section, page 11, line 299. The authors did not evaluate cell death mechanism.
11- Discussion section. The authors are repeating results already demonstrated in the manuscript. I suggest that the discussion be rewritten and that the results obtained be compared with other studies.
Author Response
Reviewer 2
“In the present study, it was done a screening of histone deacetylases inhibitors (HDACi) and microtubules inhibitors (MTi) in feline mammary tumor cell lines. Pharmacological treatment options against neoplasia in pets are scarce. The work is relevant and important to the knowledge area. However, the manuscript requires attention.
Dear reviewer, thank you so much for our remarks and corrections in order to improve the quality of our manuscript.
1- I guess that the title needs be rewritten. It is not representing the work. The work is a screening of HDACi and MTi in luminal mammary cancer cells. It is not a characterization of antitumor effects of those compounds. I suggest “Histone deacetylase inhibitors and microtubule inhibitors induce apoptosis in feline luminal mammary carcinoma cells”
We agree that the original title may not entirely represent the study. So, we accepted your suggestion to change the title to “Histone deacetylase inhibitors and microtubule inhibitors induce apoptosis in feline luminal mammary carcinoma cells”.
2- Introduction section, page 2, line 59. “HDACs have been attractive targets for…”; instead HDAC have been attractive candidates for…
Correction was performed (page 2, line 62).
3- Materials and Methods section, page 2. Did the authors perform any procedure to evaluate if the cells were contaminated with mycoplasma?
The cell lines stocks were tested for mycoplasma contamination by a PCR commercial kit (MycoSEQ™ Mycoplasma Detection Kit from Thermo Fisher). From that onwards, the cell lines were routinely inspected for any alteration of their cellular morphology and proliferation rate.
4- Materials and Methods section, page 4. Why did the authors use Histone H3 [Lys9/Lys14] as a target for acetylation? Others lysine residues as K27 (H3K27) or even other histones as H4 are important targets of HDACi in cancer control.
The HDACi target the acetylation status of different histones, however, the histone H3 is the mainly affected. H3 is primarily acetylated at Lys9, 14, 18, 23, 27, and 56, and acetylation of H3 at Lys9 appears to have a dominant role [Strahl, B. D., & Allis, C. D. (2000). The language of covalent histone modifications. Nature, 403(6765), 41–45. doi:10.1038/47412]. Others have also used these residues to evaluate the acetylation status of H3 in a similar context [Dias, J.N.R.; Aguiar, S.I.; Pereira, D.M.; André, A.S.; Gano, L.; Correia, J.D.G.; Carrapiço, B.; Rütgen, B.; Malhó, R.; Goncalves, J.; et al. The histone deacetylase inhibitor panobinostat is a potent antitumor agent in canine diffuse large B-cell lymphoma. Oncotarget 2018, 9, 28586–28598].
5- Materials and Methods section, page 4. β-actin is a constitutive cytoplasmatic protein. Why did the authors use β-actin as western blot control for histone acetylation?
Dear reviewer, we used the β-actin as a loading control, in order to normalize the different whole cell extracts, by confirming similar levels of this housekeeping protein.
6- Page 5, table 1. Table 1 must be reformatted. It is confused.
Dear review, we agree that the table 1 needed some adjustments, for a better understanding. The evaluations were split through additional columns.
7- Page 8, table 2. Table 2 must be reformatted.
As suggested, we changed the table 2 for a better reading of the IC50. The error value is now between brackets to avoid confusion.
8- Page 8, subtitle “The cytotoxic effect of HDACi is correlated with epigenetic modifications” This is not true. The authors did not perform statistical analysis that support it.
Dear reviewer, we fully agree with you. We corrected the subtitle to “The cytotoxic effect of HDACi is related with histone acetylation” (page 10, line 253).
9- Figure 3. The representative image of the western blot must be replaced. The image must represent all bands obtained in immunoblotting, without cuts or selected bands. A representative image is required for each cell line. And each graph must have properly indicated the legend of the ordinate and abscissae axis. Why did the authors analyze the results with two-tailed unpaired Student’s t-test relative to non-treated cells? Are not differences between HDACis important?
Dear reviewer, as requested the immunoblots were repeated to be presented as a single complete image for each cell line, without cuts or selected bands, and the legends were completed accordingly.
Regarding the comment of the statistical analysis, we observed some heterogeneity on the inhibitory effects of the HDACi in the different cell lines, coupled with an imperfect match between the immunoblot and apoptosis analysis. Moreover, as mentioned in the manuscript (page 7, Line 216), we could not determine an IC50 value for all drugs in all cell lines. In such cases, for the subsequent analysis (H3 acetylation status and apoptosis), we used the IC50 value obtained for the CAT-MT cell line (Table 2 of the manuscript). Indeed, we anticipated that a comparison between each cell line or even between each drug would not be accurate. We only compared for each drug in one cell line, drug-treated versus non-treated cells. Ideally, we would like to compare the effect of each drug in the different cell lines, and infer statistically whether one drug is more efficient than other in our feline cell model.
10- Discussion section, page 11, line 299. The authors did not evaluate cell death mechanism.
Dear reviewer as you suggested, the phrase was rewritten as follows: “We found that the cytotoxic effect, the mode of action and induction of apoptosis of HDACi and MTi are conserved between…” (page 16, line 314-315).
11- Discussion section. The authors are repeating results already demonstrated in the manuscript. I suggest that the discussion be rewritten and that the results obtained be compared with other studies.”
The discussion was rewritten taking into account your suggestions (e.g. some repeated results). Additionally, the results obtained were further discussed with results reported in dog, since very few studies were available concerning the use of HDACi in the cat [Dias, J.N.R.; Aguiar, S.I.; Pereira, D.M.; André, A.S.; Gano, L.; Correia, J.D.G.; Carrapiço, B.; Rütgen, B.; Malhó, R.; Goncalves, J.; et al. The histone deacetylase inhibitor panobinostat is a potent antitumor agent in canine diffuse large B-cell lymphoma. Oncotarget 2018, 9, 28586–28598].
Reviewer 3 Report
In the current manuscript, the authors have studied the effect of HDAC inhibitors on different Feline cancer cell lines. They have shown dose-dependent toxicity of both inhibitors in different cell lines. On the basis of data presented in the manuscript, these inhibitors may be potentially useful in managing Feline cancer, however more preclinical studies needed. The manuscript,s data presentation looks poor. Graphs need to be improved for better visibility. Immunoblots have been presented single-single, they should be repeated in a single gel. Furthermore, HDAC inhibitors and MTi have been shown to inhibit the growth of several cancer cell lines, previously. It will be good to study their combinatorial effect that can improve the quality of the manuscript and findings.
Author Response
Reviewer 3
“In the current manuscript, the authors have studied the effect of HDAC inhibitors on different Feline cancer cell lines. They have shown dose-dependent toxicity of both inhibitors in different cell lines. On the basis of data presented in the manuscript, these inhibitors may be potentially useful in managing Feline cancer, however more preclinical studies needed. The manuscript,s data presentation looks poor. Graphs need to be improved for better visibility. Immunoblots have been presented single-single, they should be repeated in a single gel. Furthermore, HDAC inhibitors and MTi have been shown to inhibit the growth of several cancer cell lines, previously. It will be good to study their combinatorial effect that can improve the quality of the manuscript and findings.”
Dear review, thank you so much for your remarks and corrections in order to improve the final quality of our manuscript. Concerning data presentation, the graphs were reformatted and amplified for a better visibility, whereas the immunoblots were repeated to be presented as a single complete image for each cell line, without cuts or selected bands. Additionally, new Figures 2 and 4 were uploaded with more definition (300 dpi). Regarding the combinatorial effect of HDACi and MTi, it is a great suggestion, however, without any further preclinical tests that could help us reduce the list of drugs to test, there are many possibilities to account for and, unfortunately, we don’t have the capacity to perform novel experiments due to lack of funding. Finally, in response to your observation about the preclinical studies, we made initial experiments in mice (KO Notch4 receptor, Prof. António Duarte, FMV-ULisboa). Our objective was to implant the feline cell lines in mice, as xenografts, to provide a more substantial in vivo validation of our findings. Unfortunately, the results were not satisfactory and we could not continue is this approach. It is something to revisit in the future, but not possible for now.
Round 2
Reviewer 2 Report
The authors improved the presentation of the results. The manuscript became more pleasant to read. However, I still have a question regarding the use of beta actin as a western blot control protein. Changes in histone acetylation occur both in the nucleus and in the cytoplasm, but the nuclear histone acetylation has a direct impact on gene expression. Thus, the authors should have used a nuclear protein as control, such as H1, for example. For the publication, I suggest that the authors justify and refer in the manuscript the choice of beta actin as a constitutive control of protein expression.